

# Characteristics and evolution of hemagglutinin and neuraminidase genes of Influenza A(H3N2) viruses in Thailand during 2015 to 2018

Sasiprapa Anoma[1,2], Parvapan Bhattarakosol[2,3] and Ekasit Kowitdamrong[2,3]

[1] Interdisciplinary Program in Medical Microbiology, Graduated School, Chulalongkorn University, Bangkok, Thailand

[2] Center of Excellence in Applied Medical Virology, Department of Microbiology, Faculty of Medicine, Chulalongkorn University, Bangkok, Thailand

[3] Division of Virology, Department of Microbiology, Faculty of Medicine, Chulalongkorn University, Bangkok, Thailand

Corresponding author
Ekasit Kowitdamrong,
ekasit.k@chula.ac.th

## ABSTRACT

**Background.** Influenza A(H3N2) virus evolves continuously. Its hemagglutinin (HA) and neuraminidase (NA) genes have high genetic variation due to the antigenic drift. This study aimed to investigate the characteristics and evolution of HA and NA genes of the influenza A(H3N2) virus in Thailand.

**Methods.** Influenza A positive respiratory samples from 2015 to 2018 were subtyped by multiplex real-time RT-PCR. Full-length HA and NA genes from the positive samples of influenza A(H3N2) were amplified and sequenced. Phylogenetic analysis with the maximum likelihood method was used to investigate the evolution of the virus compared with the WHO-recommended influenza vaccine strain. Homology modeling and $N$-glycosylation site prediction were also performed.

**Results.** Out of 443 samples, 147 (33.18%) were A(H1N1)pdm09 and 296 (66.82%) were A(H3N2). The A(H3N2) viruses circulating in 2015 were clade 3C.2a whereas sub-clade 3C.2a1 and 3C.2a2 dominated in 2016–2017 and 2018, respectively. Amino acid substitutions were found in all antigenic sites A, B, C, D, and E of HA but the majority of the substitutions were located at antigenic sites A and B. The S245N and N329S substitutions in the NA gene affect the $N$-glycosylation. None of the mutations associated with resistance to NA inhibitors were observed. Mean evolutionary rates of the HA and NA genes were $3.47 \times 10^{-3}$ and $2.98 \times 10^{-3}$ substitutions per site per year.

**Conclusion.** The influenza A(H3N2) virus is very genetically diverse and is always evolving to evade host defenses. The HA and NA gene features including the evolutionary rate of the influenza A(H3N2) viruses that were circulating in Thailand between 2015 and 2018 are described. This information is useful for monitoring the genetic characteristics and evolution in HA and NA genes of influenza A(H3N2) virus in Thailand which is crucial for predicting the influenza vaccine strains resulting in high vaccine effectiveness.

## INTRODUCTION

Influenza is a major cause of illness and death in humans (*Webster et al., 2013*). The seasonal outbreaks and sporadic global outbreaks of this disease result in a significant influx of patients requiring hospitalization and a substantial number of fatalities (*Hutchinson & Yamauchi, 2018*; *Lee, Jang & Seong, 2017*). The clinical manifestations of influenza include high-grade fever, headache, myalgia, sore throat, rhinorrhea, and cough (*Yamayoshi & Kawaoka, 2019*). Most people can recover from these symptoms within a week without necessitating medical intervention. However, persons who are at higher risk have the potential to develop severe sickness and mortality, *e.g.*, pregnant women, children younger than five years old, the elderly, individuals with chronic medical conditions, and obese people (*Hutchinson & Yamauchi, 2018*).

Influenza viruses are classified into the Family *Orthomyxoviridae*. Their composition consists of either seven or eight segments of negative single-stranded RNA, with an envelope. Influenza viruses are divided into four types: A, B, C, and D. Only influenza A, B, and C viruses cause infections in humans (*Subbarao & Joseph, 2007*; *Webster et al., 2013*). Hemagglutinin (HA) and neuraminidase (NA), which are glycoproteins on the surface of the influenza A virus, are utilized to categorize its subtypes (*Webster et al., 1992*; *Yamayoshi & Kawaoka, 2019*). Currently, there are a total of 18 HA (H1-H18) and 11 NA (N1-N11) subtypes. Only A(H1N1)pdm09 and A(H3N2) co-circulate as human seasonal influenza. The antibodies specific to HA and NA play an important role as a protective antibody against influenza infection.

Influenza A virus infection provides a significant public health concern due to its ability to infect a wide range of hosts, resulting in significant genetic variations caused by antigenic drift and antigenic shift. The high mutation rate is approximately $10^{-3}$ substitutions per nucleotide per strand copied (s/n/r); an average of 2–3 mutations occurs in each genome replication (*Kim et al., 2018*; *Rattanaburi et al., 2022*; *Westgeest et al., 2014*). Monitoring the influenza A virus mutation is essential to update the evolution of the virus, and discover the emergence of new variants. Furthermore, genetic information of novel influenza variations is used to forecast the specific strain that will be employed in the manufacturing of the latest influenza vaccine. The influenza vaccination might become less effective if the circulating virus is different from the vaccine strain. This study aimed to investigate the mutation of HA and NA genes of the influenza A(H3N2) virus in Thailand from 2015 to 2018 compared with the vaccine strain and the other reference strains from public databases. The findings provided information on the evolution of the HA and NA genes of the influenza A(H3N2) virus circulating in Thailand.

## MATERIALS & METHODS

### Ethics statement

The study was conducted under the Declaration of Helsinki and approved by the Institutional Review Board of the Faculty of Medicine, Chulalongkorn University, Bangkok, Thailand (IRB No. 558/62), and the date of approval is September 26, 2019.

The Institutional Biosafety Committee (IBC) of the Faculty of Medicine, Chulalongkorn University, approved this study with MDCU-IBC007/2019.

## Clinical specimens

In this study, leftover respiratory samples were obtained from the Virology laboratory, Microbiology Department, King Chulalongkorn Memorial Hospital, Thai Red Cross, Bangkok, Thailand. In accordance with IRB regulations, the hospital director's permission was given without the patient's consent, which is a prerequisite for IRB approval. A total of 443 leftover respiratory samples that tested positive for Influenza A by QuickNavi Flu+RSV (Denka Seiken, Japan) from 2015 to 2018 and stored at −80 °C after testing were enrolled in this study.

## Determination of influenza A virus subtype by multiplex real-time RT-PCR

The protocol for subtyping the influenza A virus into A(H1N1)pdm09 and A(H3N2) including the primers and probes was developed and validated in our laboratory. Viral RNA was extracted from 140 ml of the patient's respiratory samples using the QIAamp Viral RNA Mini Kit (Qiagen, Hilden, Germany) following the manufacturer's instructions. For the synthesis of complementary DNA (cDNA), 5 ml of extracted viral RNA was converted into cDNA using the Transcriptor First-Strand cDNA synthesis kit (Roche, Basel, Switzerland) and Uni12 primer (5′-AGCRAAAGCAGG-3′). Subsequently, multiplex real-time PCR was performed using the Luna Universal Probe qPCR Master Mix (New England Biolabs, Ipswich, MA, USA). The sequences of primers and hydrolysis probes were as follows: A(H1N1)pdm09-F (5′ATAYTRAGAACWCAAGAGTCTGAATG 3′), A(H1N1)pdm09-R (5′CCATGCCARTTRTCYCTG 3′), A(H1N1)pdm09-P (5′FAM-CACTAGAATCAGGRTAACAGGAGCA-BHQ1 3′), A(H3N2)-F (5′ ACAGGATTTG-CACCTTTYTC 3′), A(H3N2)-R (5′ GGAACACCYAAYTCATTCATC 3′), A(H3N2)-P (5′ HEX-ACCTTATGTGTCATGCGA-BHQ1 3′). Briefly, 5 µl of viral cDNA was added to 20 µl of total reaction volume containing 1x Luna Universal Probe qPCR Master Mix, 0.2 µM of A(H3N2) primers and probe, 0.4 µM of A(H1N1)pdm09 primers, and 0.1 µM of A(H1N1)pdm09 probe. The PCR reaction was done on the QuantStudio 5 Real-Time PCR System (Applied Biosystems, USA). The cycle steps of amplification were as follows: initial denaturation at 95 °C for 60 s, followed by 45 cycles of denaturation at 95 °C for 15 s, and extension at 60 °C for 30 s. Influenza A(H1N1)pdm09 was detected in the fluorescein (FAM; 530 nm) channel, while A(H3N2) was detected in the hexachlorofluorescein (HEX; 560 nm).

## Sequencing of Hemagglutinin and Neuraminidase genes

Viral RNA was extracted and converted into cDNA as described in the previous step. The amplification and sequencing primers were followed in the previous study (*Lee et al., 2013*). The Platinum Taq DNA Polymerase High Fidelity (Invitrogen, Waltham, MA, USA) was used to amplify the HA and NA genes following the manufacturer's instructions. The cycle steps of PCR amplification were adapted as follows: 2 cycles of denaturation at 95 °C for 5 min, annealing at 47 °C for 30 s, and extension at 68 °C for 1.5 min; 48 cycles of

95 °C for 15 s, at 58 °C (for NA gene) or 61 °C (for HA gene) for 30 s, and at 68 °C for 1.5 min, followed by the final extension at 68 °C for 10 min. All PCR steps were performed using the ProFlex PCR System (Applied Biosystems, Foster City, CA, USA). The amplified product was checked with 1.0% agarose gel electrophoresis (Affymetrix, Santa Clara, CA, USA) and purified using ExoSAP-IT PCR Product Cleanup Reagent (Applied Biosystems) according to the manufacturer's recommendations. The sequences of the HA and NA genes of influenza A(H3N2) viruses were obtained by bidirectional Sanger sequencing (Bio Basic, Markham, Ontario, Canada). The BioEdit program version 7.2, was used to analyze the data. The nucleotide sequences of the viral HA and NA genes obtained from sequencing were submitted to the GenBank database.

## Phylogenetic analysis

The phylogenetic trees of full-length HA and NA genes of influenza A(H3N2) viruses found in this study and the reference strains obtained from the GenBank and Global Initiative on Sharing Avian Influenza Data (GISAID) databases were constructed using the MEGA X program (*Kumar et al., 2018*). The Maximum Likelihood method was employed to construct the phylogenetic trees using 1,000 bootstrap replications. The Tamura 3-parameter and Gamma distributed (T92 + G) substitution model was used for the NA gene, while the Hasegawa-Kishino-Yano and Gamma distributed (HKY + G) substitution model was used for the HA gene.

## The analysis of similarity/identity between the circulating viruses and the vaccine strains

The Matrix Global Alignment Tool (MatGAT v2.01) application (*Campanella, Bitincka & Smalley, 2003*) to determine the identity or similarity between the HA and NA genes of circulating A(H3N2) viruses and the vaccine strains in the relevant year. The BLOcks Substitution Matrix (BLOSUM) 62 was selected for the scoring matrix and aligned with the sequences.

## Homology modeling of protein structure

The hemagglutinin (HA) structure of influenza A(H3N2) was generated utilizing SWISS-MODEL (https://swissmodel.expasy.org) (*Waterhouse et al., 2018*) along with the template model A/Victoria/361/2011(H3N2) (PDB ID: 4WE8). In a three-dimensional (3D) structure, the amino acid substitutions at the antigenic sites of HA proteins were illustrated utilizing the PyMOL Molecular Graphics System version 2.5.4 (*Schrödinger, 2015*). The predicted antigenic sites were obtained from previous studies (*Popova et al., 2012*; *Wiley, Wilson & Skehel, 1981*; *Wilson, Skehel & Wiley, 1981*; *Wu et al., 2020*).

## *N*-glycosylation sites prediction

Utilizing the NetNGlyc-1.0 server (*Gupta & Brunak, 2002*), the *N*-glycosylation sites were predicted. All complete amino acid sequences of HA and NA from circulating A(H3N2) viruses and A/Victoria/361/2011 (H3N2; KM821347, KJ942682) were uploaded to the server. With a potential threshold higher than 0.5, asparagine occurring in the Asn-X-Ser/Thr sequences, where X can represent any amino acid except proline, was predicted to be *N*-glycosylated.

### Evolutionary analysis and estimation of the influenza A(H3N2) virus substitution rate

Bayesian evolutionary analysis of molecular sequences utilizing Markov chain Monte Carlo (MCMC) in the Bayesian Evolutionary Analysis Sampling Trees (BEAST) program v.2.7.4 (*Bouckaert et al., 2019*) was used to estimate the evolutionary rates of the HA and NA genes. The evolutionary rate was analyzed using the birth-death skyline model (BDSKY) with the Hasegawa–Kishino–Yano (HKY) substitution model, which was determined using the maximum likelihood best-fit substitution model (ML) calculation in the MEGA X program (*Kumar et al., 2018*). The BEAST2 program was run with a strict clock model for 10 million MCMC chains and sampled every 1,000 frequencies. The substitutions per site per year were employed to measure the evolutionary rates. The prior distribution was calculated by estimating the evolution of the influenza A(H3N2) virus using the substitution rate of $4.84 \times 10^{-3}$ substitutions per site per year (*Jenkins et al., 2002*; *Phyu et al., 2022*; *Westgeest et al., 2014*). Tracer v.1.7.2 (*Rambaut et al., 2018*) was used to represent the estimated evolution parameters derived from BEAST.

### Prediction of vaccine effectiveness against influenza A(H3N2)

The predicted vaccine effectiveness (VE) was calculated using pEpitope calculator (version 2.1) (*Bonomo, Kim & Deem, 2019*) in the MATLAB program (*The MathWorks Inc., 2022*). The deduced amino acid sequences of the HA gene were submitted to measure the antigenic distance between the circulating viruses and the vaccine strains in the relevant years. The equation of the VE $= (-3.32 \times p_{epitope} + 0.66) \times 100\%$, $p_{epitope}$ is defined as the ratio of the difference between the dominant epitopes of the vaccine strain and the circulating viruses.

### Statistical analysis

GraphPad Prism version 10.0.3 (GraphPad Software, La Jolla, CA, USA) was utilized to compute the significant difference values for the vaccine effectiveness through the use of the unpaired $t$-test followed by the Mann–Whitney test. The results are displayed as the mean $\pm$ standard deviation (SD). The $p$-value was calculated at a significance level of <0.05.

## RESULTS

### Determination of influenza A virus subtype in clinical samples

To determine the subtype of influenza A virus, a total of 443 samples that tested positive for influenza A viruses were analyzed using multiplex real-time RT-PCR. 147 samples (33.18%) of influenza A(H1N1)pdm09 viruses were found. Concurrently, influenza A(H3N2) viruses were found in 296 samples (66.82%), as indicated in Table 1. Both subtypes, influenza A(H3N2) and A(H1N1)pdm09, were present at the same time. However, influenza A(H3N2) was more prevalent than A(H1N1)pdm09 in each year from 2015 to 2018 (see Fig. S1). No simultaneous infections of both subtypes were identified. It is noteworthy that a high prevalence occurs from the rainy season (June-September) through the winter season (October-January) on an annual basis. Regrettably, recruitment efforts for the samples conducted in 2017 from September to December were unsuccessful.

**Table 1  Influenza A virus subtyping by multiplex real-time RT-PCR.**

| Influenza virus | 2015 | 2016 | 2017 | 2018 | Total |
|---|---|---|---|---|---|
| A(H1N1)pdm09 | 8 (10%) | 72 (38.5%) | 17 (43.59%) | 50 (36.5%) | 147 (33.18%) |
| A(H3N2) | 72 (90%) | 115 (61.5%) | 22 (56.41%) | 87 (63.5%) | 296 (66.82%) |

**Table 2  The number of samples with HA and NA gene sequences.**

| Gene | Full-length | | | | Full-length samples/Total samples | Fragment (Sequenced samples/ Total samples) | | Complete HA+NA samples/Total samples |
|---|---|---|---|---|---|---|---|---|
| | 2015 | 2016 | 2017 | 2018 | | A | B | |
| Hemagglutinin (HA) | 7 (20%) | 14 (40%) | 4 (11.43%) | 10 (28.57%) | 35/65 (53.85%) | 38/65 (58.46%) | 50/65 (76.92%) | 26/65 (40%) |
| Neuraminidase (NA) | 1 (2.86%) | 20 (57.14%) | 3 (8.57%) | 11 (31.43%) | 35/65 (53.85%) | 36/65 (55.38%) | 41/65 (63.08%) | |

## Phylogenetic analysis of the influenza A(H3N2) viruses

To further characterize the viral HA and NA genes using bidirectional Sanger sequencing, influenza A(H3N2) positive respiratory samples were used. As a representative, two samples were chosen from each month in 2015, 2016, and 2018, while only one sample was obtained each month in 2017 because of the restricted sample size. The selected criterion for representative samples was mainly based on the Ct cycle (less than or equal to 28) from the subtyping influenza A viruses by multiplex real-time RT-PCR experiment. Therefore, a total of 65 samples were recruited from 296 influenza A(H3N2) virus samples (16 (2015), 21 (2016), 21 (2018), and 7 (2017)).

The complete sequence of the HA and NA genes was obtained from two distinct fragments (A and B), as reported in the study by Lee et al. Our study successfully obtained 35 samples (53.85%) for the entire HA and NA genes received 35 samples (53.85%). Altogether, 26 samples (40%), successfully obtained complete HA and NA gene sequences (Table 2). Together with the influenza A(H3N2) virus vaccine strains recommended by the WHO from 2010 to 2020, the sequences of the HA and NA genes from the influenza A(H3N2) positive samples were analyzed. Phylogenetic trees have been constructed for the HA (Fig. 1) and NA (Fig. 2) genes.

The A/Switzerland/9715293/2013 vaccine strain, which was categorized as clade 3C.3a, was distinct from all other influenza A(H3N2) viruses that were circulating in 2015. Nevertheless, they were all members of clade 3C.2a. The vaccine strain used in 2018, A/Singapore/INFIMH-16-0019/2016, was closely related to the circulating viruses in 2016 and 2017, which were classified under sub-clade 3C.2a1. On the other hand, the vaccine strain used in 2016–2017, A/Hong Kong/4801/2014, belonged to clade 3C.2a. 2018 revealed the discovery of two influenza A(H3N2) sub-clades, 3C.2a1b and 3C.2a2. Sub-clade 3C.2a2 was in the same clade as the vaccination strain used in 2019, A/Switzerland/8060/2017, whereas sub-clade 3C.2a1b was closely similar to the vaccine strain used in 2018 (Figs. 1 and 2).

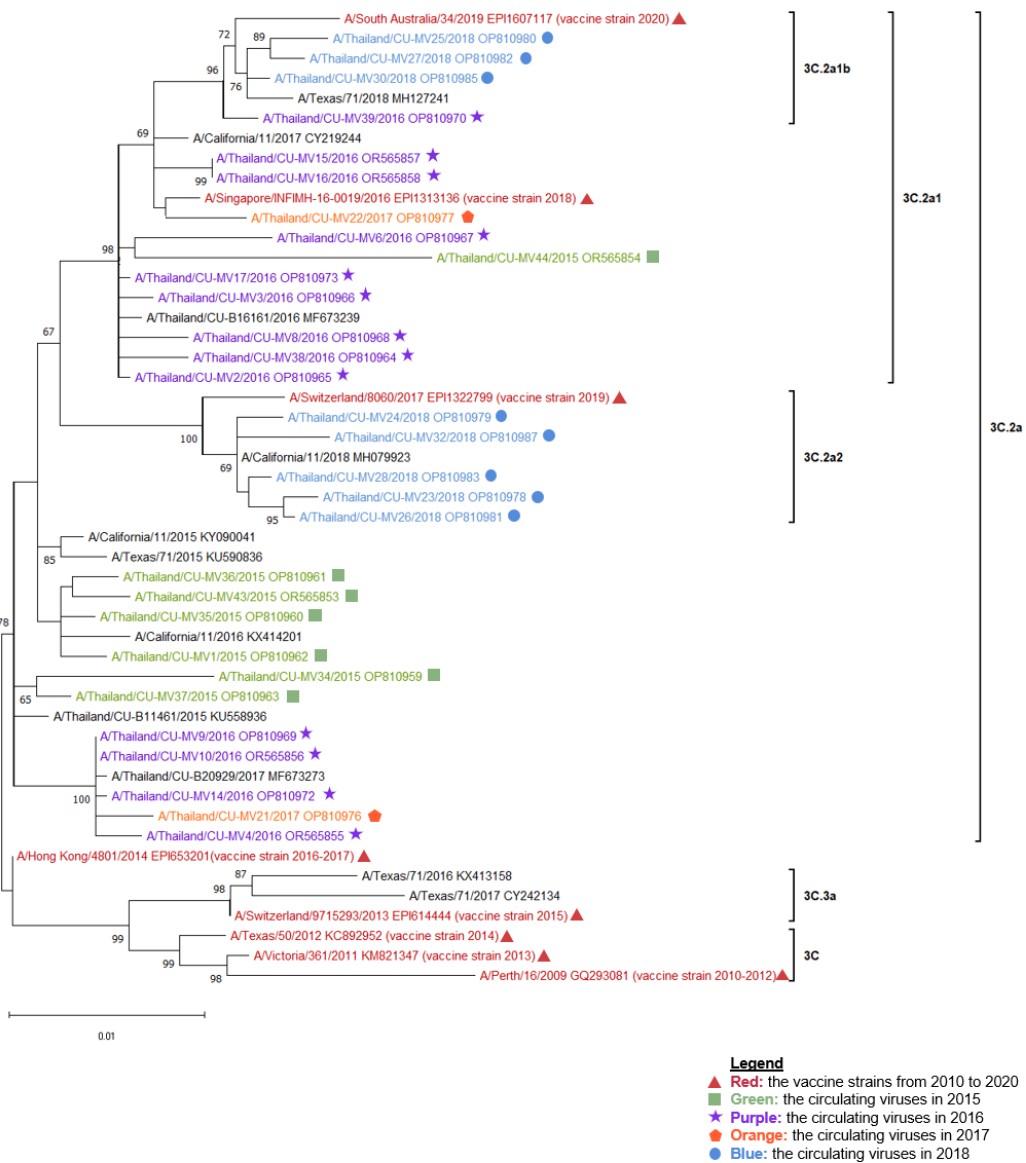

**Figure 1** **The phylogenetic analysis of the hemagglutinin (HA) gene.** The phylogenetic tree of the HA gene of the circulating influenza A(H3N2) viruses from 2015 to 2018 and the vaccine strains was constructed using the maximum likelihood method based on the Hasegawa-Kishino-Yano and Gamma distributed (HKY + G) substitution model with 1,000 bootstrap replication. Red represented the vaccine strains in the southern hemisphere from 2010 to 2020. Green represented the circulating viruses in 2015, purple; 2016, orange; 2017, and blue; 2018.

Moreover, to investigate the relationship between the A(H3N2) virus in Thailand and the global influenza A(H3N2) virus. The global influenza strains from 2015 to 2018 were compiled into a phylogenetic tree together with the HA gene of the circulating viruses, based on the continents of Africa, Asia, Europe, North America, South America, and Oceania. As shown in Fig. S2, the results showed that each year, the viruses in Thailand were closely related to viruses found in Asia, North America, and Oceania.

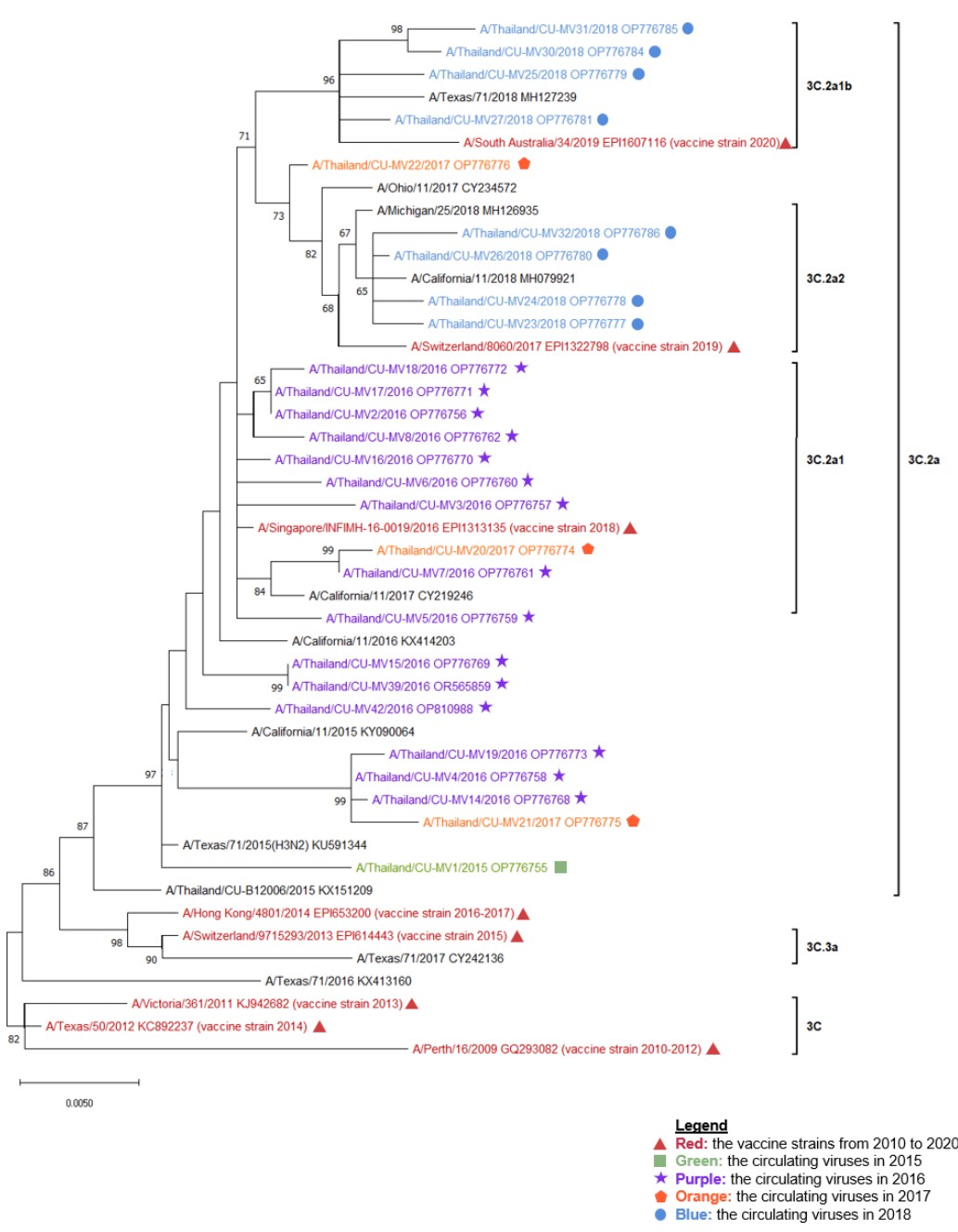

**Figure 2** **The phylogenetic analysis of the neuraminidase (NA) gene.** The phylogenetic tree of the neuraminidase (NA) gene of the circulating influenza A(H3N2) viruses from 2015 to 2018 and the vaccine strains was constructed using the maximum likelihood method based on the Tamura 3-parameter and Gamma distributed (T92 + G) substitution model with 1,000 bootstrap replication. Red represented the vaccine strains in the southern hemisphere from 2010 to 2020. Green represented the circulating viruses in 2015, purple; 2016, orange; 2017, and blue; 2018.

## Genetic variations in the HA and NA genes

Based on nucleotide sequencing, 35 sample sequences of each HA and NA gene were further analyzed. The nucleotide alterations encompassed both synonymous and non-synonymous mutations. Therefore, only non-synonymous mutations at the HA antigenic sites were extensively examined.

The MatGAT program was utilized to compute the amino acid similarities by comparing the deduced amino acid sequences of the HA gene found in circulating viruses and vaccine strains from the corresponding years. The outcomes for the years 2015, 2016, 2017, and 2018 were 97.5%, 98.4%, 98.8%, and 98.5%, respectively. HA antigenic site substitutions were identified at antigenic sites A, B, C, D, and E (Table 3). In 2015, amino acid substitutions were observed at all antigenic sites on the viruses, including A138S, R142G, and S144N (A); T128A, Y159S, and T160K (B); H311Q (C); S248T and K171N (D); and D62E and R80Q (E). However, two additional mutations were identified on antigenic sites A (T133N) and B (R197Q) in one circulating strain (A/Thailand/CU-MV34/2015). For 2016, most of the viruses exhibited the following mutations at the antigenic sites: S160K at site A; N137K and N187K at site D. The dominant amino acid substitutions identified in the 2017 viruses were antigenic site A (T151K) and antigenic site D (T137K and T187K). The viruses that exhibited the most prevalent mutations in 2018 were located at antigenic site A (T147K and G158K), antigenic site D (N137K, N187K, and A228T), and antigenic site E (E78K and R277Q). Due to most of the amino acid substitutions being changed in the amino acid group, the HA protein structure was observed. Homology modeling was used to construct the three-dimensional HA protein structure. As a constructed HA model, the influenza A/Victoria/361/2011(H3N2) virus was utilized for demonstrating the antigenic sites A–E. (Figs. 3A and 3B). The constructed HA model was substituted with mutant HA sites that have been identified in circulating viruses. In our study, the mutations seem improbable to affect the tertiary structure of the HA protein (Figs. 3C–3F).

The amino acid similarities of the NA gene were determined similar to the HA gene. The findings indicated significant resemblances of 98.7% (2015), 99.1% (2016), 98.3% (2017), and 98.9% (2018). The amount of amino acid substitutions in 2015, 2016, 2017, and 2018 were 10, 12, 18, and 7 positions, respectively (Table 4).

## The prediction of *N*-glycosylation site in the HA and NA genes

Glycosylation is an important post-translational modification that significantly impacts viruses. Substitutions of amino acids may affect the glycosylation pattern. Therefore, the *N*-glycosylation sites located in the HA gene have the potential to impact not only the antigenicity of the HA protein but also the viral infectivity efficacy. The *N*-glycosylation sites were predicted by comparing whole amino acid sequences from 35 circulating viruses to those of the A/Victoria/361/2011. The findings revealed that all circulating viruses had one additional *N*-glycosylation site (N174) and lacked N160, in contrast to the thirteen *N*-glycosylation sites found in the HA gene of A/Victoria/361/2011 (Table 5). However, three viruses were identified that possessed distinctive *N*-glycosylation locations. A/Thailand/CU-MV34/2015 possessed one more additional *N*-glycosylation site (N112) and lacked N149. A/Thailand/CU-MV22/2017 lacked the *N*-glycosylation site N149. Lastly,

Anoma et al. (2024), PeerJ, DOI 10.7717/peerj.17523

**Table 3 Amino acid substitutions on antigenic sites in the HA gene of the circulating viruses from 2015 to 2018 compared with vaccine strains.**

| Year | Vaccine strain | Circulating strain | Amino acid substitution on antigenic sites | | | | |
|---|---|---|---|---|---|---|---|
| | | | A | B | C | D | E |
| 2015 | A/Switzerland/9715293/2013 EPI614444 | A/Thailand/CU-MV1/2015 OP810962 | S154A, G158R, *N160S | A144T, S175Y, K176T | Q327H | | |
| | | A/Thailand/CU-MV34/2015 OP810959 | N149T, S154A, G158K, *N160S | A144T, S175Y, K176T, Q213R | Q327H | | E78D |
| | | A/Thailand/CU-MV35/2015 OP810960 | S154A, G158R, *N160S | A144T, S175Y, K176T | Q327H | T264S | |
| | | A/Thailand/CU-MV36/2015 OP810961 | S154A, G158R, *N160S | A144T, S175Y, K176T | Q327H | | |
| | | A/Thailand/CU-MV37/2015 OP810963 | S154A, G158K, *N160S | A144T, S175Y, K176T | Q327H | T264S | |
| | | A/Thailand/CU-MV43/2015 OR565853 | S154A, G158R, *N160S | A144T, S175Y, K176T | Q327H | | |
| | | A/Thailand/CU-MV44/2015 OR565854 | S154A, G158R, *N160S | A144T, S175Y, K176T | Q327H | *N187K | Q96R |
| 2016 | A/Hong Kong/4801/2014 EPI653201 | A/Thailand/CU-MV2/2016 OP810965 | | | | *N187K | |
| | | A/Thailand/CU-MV3/2016 OP810966 | | | | *N187K, S263N | |
| | | A/Thailand/CU-MV4/2016 OR565855 | *S160K | | | *N137K | |
| | | A/Thailand/CU-MV6/2016 OP810967 | T151K, R158K | F209S | | *N187K | |
| | | A/Thailand/CU-MV8/2016 OP810968 | | | | *N187K, S235Y | |
| | | A/Thailand/CU-MV9/2016 OP810969 | *S160K | | | *N137K | |
| | | A/Thailand/CU-MV10/2016 OR565856 | *S160K | | | *N137K | |
| | | A/Thailand/CU-MV13/2016 OP810971 | *S160K | | | *N137K | |
| | | A/Thailand/CU-MV14/2016 OP810972 | *S160K | | | *N137K | |
| | | A/Thailand/CU-MV15/2016 OR565857 | R158G | | | *N137K, *N187K | |
| | | A/Thailand/CU-MV16/2016 OR565858 | R158G | | | *N137K, *N187K | |
| | | A/Thailand/CU-MV17/2016 OP810973 | | | | *N187K | |
| | | A/Thailand/CU-MV38/2016 OP810964 | | | | *N187K, I230V | |
| | | A/Thailand/CU-MV39/2016 OP810970 | | | H327Q | *N137K, *N187K | K108R |
| 2017 | A/Hong Kong/4801/2014 EPI653201 | A/Thailand/CU-MV21/2017 OP810976 | N138D, *S160K | | | *N137K | |
| | | A/Thailand/CU-MV22/2017 OP810977 | *T151K | F209S | | *N137K, *N187K | |
| | | A/Thailand/CU-MV40/2017 OP810974 | *T151K | F209S | | *N137K, *N187K | |
| | | A/Thailand/CU-MV41/2017 OP810975 | *T151K | F209S | | *N137K, *N187K | |

**Table 3** (*continued*)

| Year | Vaccine strain | Circulating strain | Amino acid substitution on antigenic sites | | | | |
|------|----------------|-------------------|-----|-----|-----|-----|-----|
| | | | A | B | C | D | E |
| 2018 | A/Singapore/INFIMH-16-0019/2016 EPI1313135 | A/Thailand/CU-MV23/2018 OP810978 | T147K, G158K | | | *K137N, *K187N, A228T | E78K, R277Q |
| | | A/Thailand/CU-MV24/2018 OP810979 | T147K, G158K | | | *K137N, *K187N, A228T, V229A | R277Q |
| | | A/Thailand/CU-MV25/2018 OP810980 | *T151K | T144A | I64R, V325I, H327Q | | E78G, K108R |
| | | A/Thailand/CU-MV26/2018 OP810981 | T147K, G158K | | | *K137N, *K187N, A228T | E78K, R277Q |
| | | A/Thailand/CU-MV27/2018 OP810982 | *T151K | T144A, S214P | V325I, H327Q | | E78G, K108R |
| | | A/Thailand/CU-MV28/2018 OP810983 | T147K, G158K | | D69N | *K137N, *K187N, A228T | E78K, R277Q |
| | | A/Thailand/CU-MV29/2018 OP810984 | T147K, G158K | | | *K137N, *K187N, A228T | E78K, R277Q |
| | | A/Thailand/CU-MV30/2018 OP810985 | *T151K | | Q327H | | E78G, K108R |
| | | A/Thailand/CU-MV31/2018 OP810986 | *T151K | | Q327H | | E78G, K108R |
| | | A/Thailand/CU-MV32/2018 OP810987 | T147K, G158K | | | *K137N, *K187N, A228T | R277Q |

**Notes.**

*Represented the positions that may affect the function properties of the proteins.

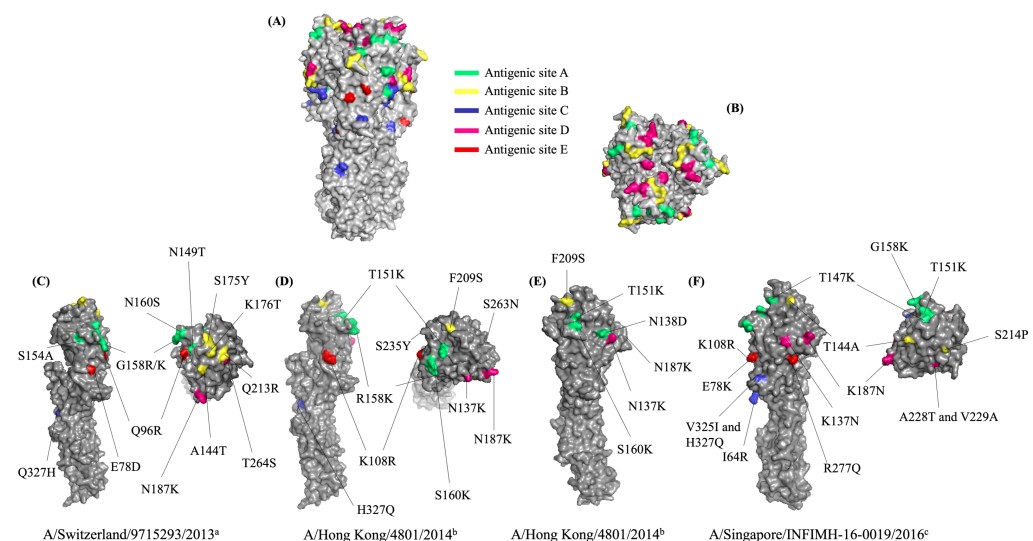

**Figure 3** **The three-dimensional protein structure of the Hemagglutinin (HA) of the influenza A(H3N2) virus.** The three-dimensional protein structure of the Hemagglutinin (HA) of the influenza A(H3N2) virus. (A) Side view and (B) Top view of the HA demonstrating the antigenic sites A-E, green color represents the antigenic site A, yellow color represents the antigenic site B, blue color represents the antigenic site C, pink color represents the antigenic site D, and red color represents the antigenic site E. (C) The amino acid substitutions found at the antigenic sites in the circulating influenza A(H3N2) viruses from 2015, (D) from 2016, (E) from 2017, and (F) from 2018. a WHO recommended influenza vaccine strain in 2015 A/Switzerland/9715293/2013, b vaccine strain in 2016 and 2017 A/Hong Kong/4801/2014, and c vaccine strain in 2018 A/Singapore/INFIMH-16-0019/2016.

two $N$-glycosylation sites (N142 and N149) were removed in A/Thailand/CU-MV27/2018. At antigenic site B of the HA gene, a non-synonymous mutation (N160S) was found, which is in line with the outcomes of the prior experiment. This mutation potentially contributes to the absence of the $N$-glycosylation site N160.

In the NA of A/Victoria/361/2011, seven predicted $N$-glycosylation sites were identified. Two additional $N$-glycosylation sites (N245 and N329) were identified in all influenza A(H3N2) viruses in Thailand (Table 6). Correlating with the previous result, the non-synonymous mutations in the NA gene (S245N and N329S) were identified. These mutations may have the potential to impact the addition of $N$-glycosylation sites N245 and N329.

## The evolution rate of influenza A(H3N2) virus

To ascertain the influenza A(H3N2) virus's evolutionary rate in the HA and NA genes, the Bayesian MCMC approach was used to estimate the evolutionary rate. In the case of the HA gene, the mean evolutionary rate was $3.47 \times 10^{-3}$ substitutions per site per year with 95% highest posterior density (HPD) between $2.55 \times 10^{-3}$ to $4.43 \times 10^{-3}$ substitutions per site per year. For the NA gene, the mean evolutionary rate was $2.98 \times 10^{-3}$ substitutions per site per year with 95% HPD between $2.17 \times 10^{-3}$ to $3.90 \times 10^{-3}$ substitutions per site per year.

**Table 4  Amino acid substitutions in the NA gene of the circulating viruses from 2015 to 2018 compared with vaccine strains.**

| Influenza A(H3N2) viruses | Amino acid substitutions | | | | | | | | | | | |
|---|---|---|---|---|---|---|---|---|---|---|---|---|
| **2015** | **37** | **47** | **48** | **53** | **215** | ***245** | ***247** | **267** | **380** | **392** | | |
| A/Switzerland/9715293/2013[a] | F | N | N | C | V | S | S | K | I | T | | |
| A/Thailand/CU-MV1/2015 | I | K | T | V | I | N | T | T | V | I | | |
| **2016** | **93** | **140** | **161** | **231** | ***245** | ***247** | **267** | **339** | **380** | **392** | **409** | ***468** |
| A/Hong Kong/4801/2014[b] | G | L | N | I | S | S | T | D | I | T | I | P |
| A/Thailand/CU-MV2/2016 | G | I | N | V | N | T | K | N | V | I | V | H |
| A/Thailand/CU-MV4/2016 | D | L | S | V | N | T | K | N | V | I | I | L |
| **2017** | **93** | **127** | **161** | **176** | **231** | ***245** | ***247** | **267** | **286** | **315** | ***329** | **339** |
| A/Hong Kong/4801/2014[b] | G | D | N | I | I | S | S | T | G | S | N | D |
| A/Thailand/CU-MV20/2017 | G | V | N | I | V | N | T | K | G | R | N | N |
| A/Thailand/CU-MV21/2017 | D | D | S | I | V | N | T | K | D | S | N | N |
| A/Thailand/CU-MV22/2017 | G | D | N | M | V | N | T | K | G | S | S | N |
| **2017** | **346** | **368** | **380** | **392** | **401** | ***468** | | | | | | |
| A/Hong Kong/4801/2014[b] | G | E | I | T | G | P | | | | | | |
| A/Thailand/CU-MV20/2017 | G | E | V | I | D | R | | | | | | |
| A/Thailand/CU-MV21/2017 | V | E | V | I | G | L | | | | | | |
| A/Thailand/CU-MV22/2017 | G | K | V | I | G | H | | | | | | |
| **2018** | **126** | **176** | **212** | **220** | ***303** | ***329** | **386** | | | | | |
| A/Singapore/INFIMH-16-0019/2016[c] | P | I | I | K | V | N | P | | | | | |
| A/Thailand/CU-MV23/2018 | P | M | V | K | V | S | S | | | | | |
| A/Thailand/CU-MV30/2018 | L | I | V | N | I | S | P | | | | | |

**Notes.**
[a]The WHO recommended influenza vaccine strain in 2015.
[b]The WHO recommended influenza vaccine strain in 2016 and 2017.
[c]The WHO recommended influenza vaccine strain in 2018.
*Represented the position previously reported to be involved in host immune response.

## The predicted vaccine effectiveness against influenza A(H3N2) from 2015 to 2018

According to the results of our phylogenetic analysis and genetic variations, vaccine protection may be affected by mutations occurring on the epitope of the HA gene. The pEpitope calculator was utilized to compute the predicted vaccine effectiveness (VE) against the circulating influenza A(H3N2) viruses. As shown in Fig. 4, the mean $p_{epitope}$ values between the vaccine strain and the circulating viruses were 0.17, 0.05, 0.07, and 0.1 whereas the average and standard error of the predicted VE was $10.56 \pm 12.28\%$, $48.48 \pm 8.61\%$, $43.65 \pm 8.94\%$, and $31.63 \pm 9.88\%$ in 2015, 2016, 2017, and 2018, respectively. The findings revealed that the predicted VE in 2015 was statistically significantly lower than those in 2016–2018.

## DISCUSSION

This study involved the recruitment of 443 influenza A-positive samples in Thailand from 2015 to 2018. The purpose was to subtype the influenza A viruses using multiplex real-time RT-PCR. Both influenza A(H1N1)pdm09 and A(H3N2) viruses were observed

**Table 5  The predicted N-glycosylation sites in the HA gene.**

| N-glycosylation sites | Influenza A(H3N2) viruses | | | | |
|---|---|---|---|---|---|
| | A/Victoria /361/2011 | A/Thailand /CU-MV34/2015 | A/Thailand /CU-MV14/2016 | A/Thailand /CU-MV22/2017 | A/Thailand /CU-MV27/2018 |
| 24 NSTA | + | + | + | + | + |
| 38 NGTI | + | + | + | + | + |
| 54 NATE | + | + | + | + | + |
| 61 NSSI | + | + | + | + | + |
| 79 NCTL | + | + | + | + | + |
| *112 NCSP | − | + | − | − | − |
| 138 NESF | + | + | + | + | + |
| *142 NWTG | + | + | + | + | − |
| *149 NGTS | + | − | + | − | − |
| *160 NNSF | + | − | − | − | − |
| *174 NYTY | − | + | + | + | + |
| 181 NVTM | + | + | + | + | + |
| 262 NSTG | + | + | + | + | + |
| 301 NGSI | + | + | + | + | + |
| 499 NGTY | + | + | + | + | + |

Notes.
*The different N-glycosylation sites among the circulating influenza A(H3N2) viruses and the A/Victoria/361/2011.

**Table 6  The predicted N-glycosylation sites in the NA gene.**

| N-glycosylation sites | Influenza A(H3N2) viruses | | | | |
|---|---|---|---|---|---|
| | A/Victoria/ 361/2011 | A/Thailand/ CU-MV34/2015 | A/Thailand/ CU-MV14/2016 | A/Thailand/ CU-MV22/2017 | A/Thailand/ CU-MV27/2018 |
| 61 NITE | + | + | + | + | + |
| 70 NTTI | + | + | + | + | + |
| 86 NWSK | + | + | + | + | + |
| 146 NNTV | + | + | + | + | + |
| 200 NATA | + | + | + | + | + |
| 234 NGTC | + | + | + | + | + |
| *245 NATG | − | + | + | + | + |
| *329 NDSS | − | + | + | + | + |
| 367 NETS | + | + | + | + | + |

Notes.
*The different N-glycosylation sites among the circulating influenza A(H3N2) viruses and the A/Victoria/361/2011.

to be circulating simultaneously. Nevertheless, influenza A(H3N2) was the predominant strain over A(H1N1)pdm09 (66.82% *vs* 33.18%, Table 1), similar to the previous studies (46.7%–69.4% *vs* 8.5%–37.2%) (*Boonnak et al., 2021*; *Suntronwong et al., 2017*; *Tewawong et al., 2016*). We performed whole sequencing on both the HA and NA genes because both are utilized for subtyping influenza A virus and the HA protein is crucial for the induction of neutralizing antibodies. Only A(H3N2) positive samples were randomly selected since there was less information on A(H3N2) than A(H1N1)pdm09 (Table 2). Our

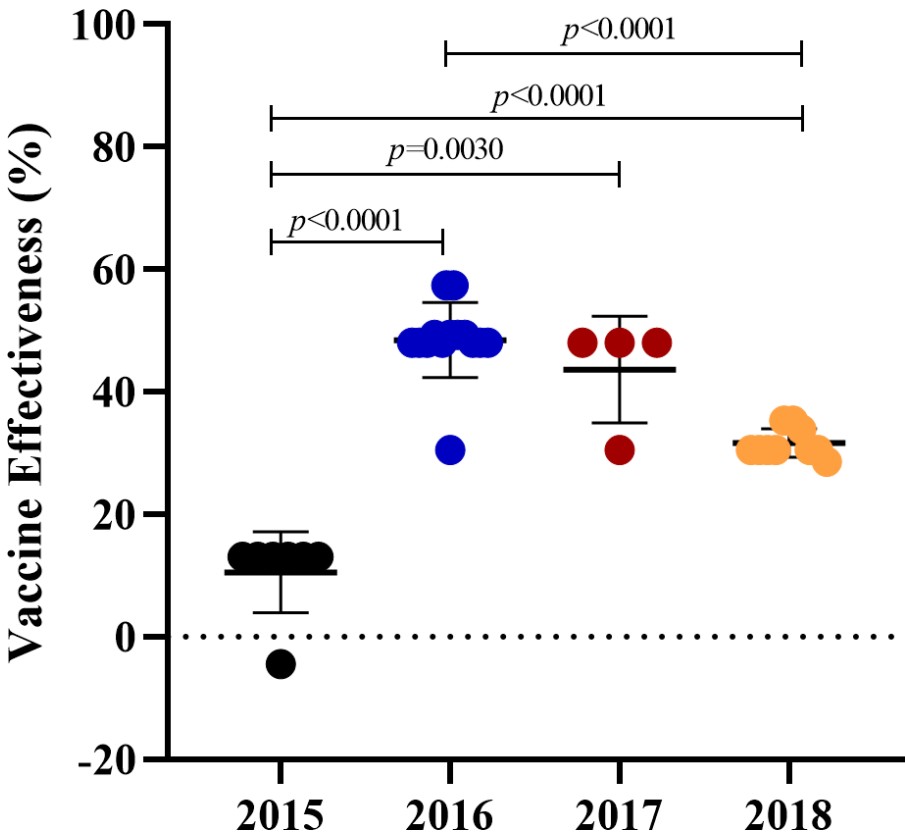

**Figure 4** **The predicted vaccine effectiveness of the A(H3N2) circulating viruses from 2015 to 2018.** The predicted vaccine effectiveness was calculated using the pEpitope Calculator. The p_epitope calculated the antigenic distance between the A(H3N2) circulating viruses and the vaccine strains (2015; A/Switzerland/9715293/2013, 2016 and 2017; A/Hong Kong/4801/2014, and 2018; A/Singapore/INFIMH-16-0019/2016). Circles represented the circulating viruses. The error bar represented SD. *p*-value indicated a significant difference (unpaired *t*-test).

results revealed sub-clade 3C.2a (2015), while sub-clade 3C.2a1, 3C.2a1b, and 3C.2a2 were identified in 2016–2018 (Figs. 1 and 2); these were the same sub-clades that were reported in Myanmar (*Phyu et al., 2022*) and another report of Thailand (*Suntronwong et al., 2021*).

As the influenza virus's HA protein has a significant genetic variation and is essential to the viral pathogenicity, it is imperative to determine its genetic diversity. An analysis was conducted to compare the genetic variation of the HA gene of circulating viruses with the vaccine strain in the relevant year. Ten amino acid substitutions were identified at all antigenic sites in the 2015 circulating viruses. During the same period, five of these substitutions were also observed in Myanmar (*Phyu et al., 2022*) and another study in Thailand (*Boonnak et al., 2021*). Our study exclusively discovered four amino acid substitutions, *i.e.,* E78D, Q96R, N149T, and T264S (Table 3). A study demonstrated an association between the fusion activation pH of HA and the F175Y/D391G mutation, which enhances the viral ability to evade the immune response during the late endosomes/lysosomes fusion step (*Chai et al., 2016*). Nevertheless, in our study, the S175Y

mutation was mutated the same as F175Y, suggesting this mutation may have a potential impact on the viral immune escape. Clade 3C.2a1 was classified using four amino acid substitutions (N137K, N187K, I422V, and G500E) out of eleven that occurred in 2016 and 2017. The substitution of N138D and T151K were reported to be an *N*-glycosylation site which affects the viral immune evasion (*Jagadesh et al., 2019*). The F209S substitution is related to the enhancement of viral replication (*Chambers et al., 2014*). Moreover, the K108R was found only in our study. In 2018, 19 amino acid substitutions were observed. Most of them were also reported in the previous studies in Thailand and Myanmar except the A228T and V325I (*Boonnak et al., 2021*; *Phyu et al., 2022*; *Rattanaburi et al., 2022*) (Table 3).

The alteration of *N*- linked glycosylation in the antigenic sites of the HA affects the antibody binding affinity (*Melidou et al., 2017*; *Phyu et al., 2022*; *Zost et al., 2017*). Typically, potential **N**-glycosylation sites can be found on an asparagine (Asn) in the sequon Asn-X-Ser/Thr where X may represent any amino acid except proline (*Allen & Ross, 2018*). By comparing the predicted *N*-glycosylation sites between the circulating viruses and A/Victoria/361/2011, all of the circulating viruses gained one additional *N*-glycosylation site (N174) but lost one (N160) (Table 5). The addition of **N**-glycosylation at N174 on the globular head of the HA might help the virus evade immunity by shielding the viral epitopes from neutralizing antibodies. The absence of N149 in the A/Thailand/CU-MV34/2015 strain may be attributed to an amino acid substitution occurring at the antigenic site A (N149T; Table 3). The loss of *N*-glycosylation site N160 which is located around the receptor binding site of the HA might result in an enhancement of viral replication (*Schulze, 1997*).

Currently, there are several neuraminidase inhibitors (NAI) that are available and recommended for the treatment of influenza A and B. These include zanamivir, oseltamivir, peramivir, and laninamivir. Mutations related to NAI resistance of influenza A(H3N2) viruses have been previously reported, such as E119V, Q136K, D151E, I222L/V, R224K, N245Y, K249E, E276D, R292K, N294S, R371K and Q391K (*Boonnak et al., 2021*; *Eshaghi et al., 2014*). Fortunately, the mutations associated with NAI resistance were not observed in the circulating influenza A(H3N2) viruses in our study. This result corresponded to the previous studies in Thailand (*Boonnak et al., 2021*; *Rattanaburi et al., 2022*). Nevertheless, V303I substitution was found in a virus circulating in 2018 (Table 4), this mutation was demonstrated to be associated with borderline resistance against oseltamivir and zanamivir (*Rattanaburi et al., 2022*; *Takashita et al., 2020*). Hence, this virus has the potential to develop NAI resistance in the future because of its rapid mutation rate.

From the *N*-glycosylation site prediction results, two additional *N*- glycosylation sites (S245N and N329S) were demonstrated (Table 6). Previous studies reported the amino acid substitutions at S245N and S247T predominantly occurred as a double mutation in the NA of clade 3C.2a (*Blumenkrantz et al., 2021*; *Sleeman et al., 2014*; *Wan et al., 2019*). Our results confirmed these studies. These two mutations encoded an *N*-glycosylation site closer to the sialidase rim and might aid the virus to evade host immunity. In addition, *N*-glycosylation site N329 was discovered in all circulating viruses. Its function involved the reduction of sialidase activity and viral immune evasion (*Hussain et al., 2021*). Moreover,

the circulating viruses have acquired P468H/L substitution since 2016 (Table 4) which might result in the reduction of the host immune response against the virus (*Allen & Ross, 2018*; *Wan et al., 2019*).

The influenza virus evolves rapidly to improve viral replication and evade the host immune response, especially in surface glycoprotein HA and NA genes by accumulating the point mutation. The evolutionary rates of the HA and NA genes of the influenza A(H3N2) virus were estimated with the Bayesian evolutionary analysis using a molecular clock. In comparison to a study from Myanmar ($3.37 \times 10^{-3}$), our mean evolutionary rate substitutions per site per year of HA ($3.47 \times 10^{-3}$) was similar (*Phyu et al., 2022*). The evolutionary rate appears to be lower than that of the influenza A(H1N1)pdm09 virus ($3.18 \times 10^{-3}$) (*Al Khatib, Al Thani & Yassine, 2018*). This could be the cause of the yearly strain variations in the influenza A(H3N2) virus used in vaccines. Our mean evolutionary rate ($2.98 \times 10^{-3}$) for the NA gene of the influenza A(H3N2) virus was also similar to that of a study from Thailand ($3.18 \times 10^{-3}$) (*Tewawong et al., 2017*) and a study from Myanmar ($2.89 \times 10^{-3}$) (*Phyu et al., 2022*). However, a study conducted between 1968 and 1995 revealed a slightly higher evolutionary rate ($2.28 \times 10^{-3}$) (*Xu et al., 1996*). Several studies confirmed higher evolutionary rates in HA and NA genes than in the other six genes (*Nobusawa & Sato, 2006*; *Parvin et al., 1986*; *Phyu et al., 2022*).

Low vaccine protection may arise from the vaccine mismatch caused by an amino acid alteration on an epitope in the HA protein. Influenza VE is usually 40–60% when the circulating viruses are well-matched to the vaccine strain used in that season (*Martins et al., 2023*; *Osterholm et al., 2012*; *Tricco et al., 2013*). The mismatch between the vaccine strain and the circulating viruses may substantially reduce the influenza VE (*Chon et al., 2019*; *Puig-Barbera et al., 2019*; *Rondy et al., 2018*). The predicted VE in 2015 was only 10.56% (SE, 12.28%), significantly lower than in other seasons (Fig. 4). It might be associated with the genetic variation outcomes of the circulating virus in 2015, where a greater number of amino acid substitutions were detected at antigenic sites A and B compared to the other years. These substitutions could potentially impact vaccine efficacy at the year of 2015. Our finding was consistent with the cohort study in the elderly in Northeast Thailand which reported the VE against the dominant influenza A(H3N2) virus was just 2% with a 95% confidence interval (CI) between -99% and 51% during the same period (*Prasert et al., 2019*). In 2016 and 2017, the predicted VE ($48.48 \times 8.61\%$ and $43.65 \times 8.94\%$) showed moderate protection against influenza A(H3N2) which related to the studies of influenza VE in the same season from Canada (36%, CI [18–50]%) (*Skowronski et al., 2017*), the United States (33%, CI [23–41]%), and Hong Kong (52.8%, CI [17.1–73.2]%) (*Flannery et al., 2019*). In 2018, the predicted VE revealed a slightly decreased protection, 31.63% (SE, 9.88%) in contrast to a study from the United States which reported VE against influenza A(H3N2) in the 2018–2019 season was 9% CI [0–20%] (*Chung et al., 2020*). The discrepancy may be due to the influenza A(H3N2) virus sub-clade 3C.3a was also found to be co-circulated in the United States during 2018–2019 which might result in a lower vaccine effectiveness (*Doyon-Plourde, Fortin & Quach, 2022*; *Flannery et al., 2020*).

Some limitations may affect our findings. First, influenza A-positive respiratory samples in this study were collected from patients only in Bangkok which might not represent the

influenza A(H3N2) viruses circulating in every area of Thailand. Second, the influenza A(H3N2) positive samples from 2015 to 2018 were randomly selected in a small sample size, due to the lower amount of samples, for further characterization of the HA and NA genes which might not reflect the true molecular epidemiology of the HA and NA of the influenza A(H3N2) viruses in Thailand. Third, this study focused only on the HA and NA genes of the influenza A(H3N2) viruses whereas the other six viral gene segments may also be involved in the evolution and genetic diversity of the influenza A(H3N2) viruses. For further studies, the sample size should be increased to ensure the investigation of the genetic variation. To predict the vaccine strain and assess the efficacy of the vaccine, it is important to note that whole genome sequencing of the HA gene is still required. Although the NA gene sequence does not play a role in vaccine development, it is crucial for predicting antiviral drug-resistant strains.

## CONCLUSIONS

This study revealed the characteristics and evolution of influenza A(H3N2) viruses in Thailand during 2015–2018. Our study found that the influenza A(H3N2) viruses predominated over A(H1N1)pdm09. The A(H3N2) viruses circulating in 2015 were clade 3C.2a, and sub-clade 3C.2a1 dominated in 2016 and 2017. Moreover, the A(H3N2) viruses circulating in 2018 were clustered in clade 3C.2a2. The amino acid substitutions of the HA gene occurred in antigenic sites A, B, C, and E which could lead to immune evasion of the viruses. In addition to the NA gene, the mutations affected the $N$-glycosylation sites, which could also result in evading host immunity. The HA and NA genes showed a high evolution rate which were $3.47 \times 10^{-3}$ and $2.98 \times 10^{-3}$ substitutions per site per year. The influenza vaccine's effectiveness in 2015 was notably lower than in other years because the viruses were not closely matched to the vaccine strain recommended by the WHO, where a higher number of substitutions of amino acids was found at antigenic sites A and B than in other years. The features and evolution of the influenza A(H3N2) viruses in Thailand in relation to vaccination efficacy can be better understood with the support of this information.

## ACKNOWLEDGEMENTS

We are grateful to the staff in the Virology Unit, Department of Microbiology, Faculty of Medicine, Chulalongkorn University for the influenza A-positive respiratory sample collection.

### Funding

This research was funded by the 100th Anniversary Chulalongkorn University for Doctoral Scholarship, CU Graduated School Thesis Grant from Chulalongkorn University and Ratchadaphiseksomphot Matching Fund (grant number RA-MF-15/65 by Parvapan Bhattarakosol), Faculty of Medicine, Chulalongkorn University. There was no additional external funding received for this study. The funders had no role in study design, data collection and analysis, decision to publish, or preparation of the manuscript.

## Grant Disclosures

The following grant information was disclosed by the authors:

Chulalongkorn University and Ratchadaphiseksomphot Matching Fund: RA-MF-15/65.

## Competing Interests

The authors declare there are no competing interests.

## Author Contributions

- Sasiprapa Anoma conceived and designed the experiments, performed the experiments, analyzed the data, prepared figures and/or tables, and approved the final draft.
- Parvapan Bhattarakosol conceived and designed the experiments, analyzed the data, authored or reviewed drafts of the article, supervision, funding acquisition, and approved the final draft.
- Ekasit Kowitdamrong conceived and designed the experiments, analyzed the data, authored or reviewed drafts of the article, supervision, project administration, funding acquisition, and approved the final draft.

## Human Ethics

The following information was supplied relating to ethical approvals (i.e., approving body and any reference numbers):

The study was conducted in accordance with the Declaration of Helsinki and received ethical approval from the Institutional Review Board of the Faculty of Medicine, Chulalongkorn University, Bangkok, Thailand on September 26, 2019 (IRB No. 558/62).

## Ethics

The following information was supplied relating to ethical approvals (i.e., approving body and any reference numbers):

The study was conducted in accordance with the Declaration of Helsinki and approved by the Institutional Review Board of the Faculty of Medicine, Chulalongkorn University, Bangkok, Thailand (IRB No. 558/62) and the date of approval is 26 September 2019. The Institutional Biosafety Committee (IBC) of the Faculty of Medicine, Chulalongkorn University approved this study with MDCU-IBC007/2019.

## DNA Deposition

The following information was supplied regarding the deposition of DNA sequences:

The Hemagglutination and Neuraminidase sequences were submitted to the GenBank database with accession numbers OP776755–OP776787, OP810959–OP810988, and OR565853–OR565859. Due to the unavailable data, the sequences were uploaded in Figs. S1 and S2.

## Data Availability

The raw data are available in the Supplemental Files.

## Supplemental Information

Supplemental information for this article can be found online at http://dx.doi.org/10.7717/peerj.17523#supplemental-information.

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
