# Peer review of "Characteristics and evolution of hemagglutinin and neuraminidase genes of Influenza A(H3N2) viruses in Thailand during 2015 to 2018"

_PeerJ, doi:10.7717/peerj.17523_

## Round 0.1 · original submission · Minor Revisions

Dear authors,

Based on the revision of the three reviewers, I decided that your manuscript requires a few Minor Revisions. Please, address specifically the questions posed by Reviewer 2 and 3 and revise your manuscript accordingly.

·

Basic reporting

no comment

Experimental design

no comment

Validity of the findings

no comment

Additional comments

This well-written article is the result of extensive work and a good understanding of the topic it deals with. I enjoyed reading it, and I gladly recommend its publishing.

·

Basic reporting

No comment.

Experimental design

For the Experimental design, it is necessary to describe whether the item "Determination of influenza A virus subtype by multiplex real-time RT-PCR" was validated by the work or cite the work on which it was based.

Validity of the findings

In line 216, summarize the text only by citing the quantity/percentage of samples with complete sequencing for NA and HA. Table 2 informing the number of samples with complete sequencing or not is of little use for the results.



In conclusion lines 449 to 453 other possibilities should be considered. Therefore, mentioning that the findings of the work, together with other variables, contribute to understanding the low efficiency of vaccination in Thailand in 2015

Additional comments

A legend in figure 1 with colors would facilitate better understanding.

·

Basic reporting

The authors analyzed the hemagglutinin (HA) and neuraminidase (NA) genetic variations of the influenza-positive samples from Flu patients in King Chulalongkorn Memorial Hospital from 2015 to 2018. They did further phylogenetic analysis, investigated the virus genetic evolution and compared the sequence with the influenza virus circulating in other geographic areas as well as WHO recommended vaccine strain. The results from this retrospective study are useful to help understand the influenza virus strains circulating and evolving in Thailand and the genetic difference from the WHO recommended vaccine strain, which will help guide the future vaccine design to improve the vaccine efficacy. The study design and methodologies well supported the study purpose. The manuscript is overall well written.

I recommend acceptance with revisions to address the following questions:
• Line 211-213: Why only two sequences from each month were chosen as representatives? The criteria for representative selection were not described and the rationale of choosing two was not mentioned. Without proper criteria, the selected representatives might not really represent the whole.
• Line 216-222: how was the percentage calculated was not explained, making data from Table 2 hard to understand. For example, what is the divisor to obtain 53.85% and 58.64%.

Experimental design

No comment

Validity of the findings

No comment

Additional comments

No comment

---

## Round 0.2 · accepted · Accept

As the authors have addressed all the reviewers' comments, and all reviewers considered that the manuscript is in its final form to be published, it is a great pleasure to announce that it has been accepted for publication! Congratulations!!!

·

Basic reporting

No further comments.

Experimental design

No further comments.

Validity of the findings

No further comments.

Additional comments

No further comments.